# How Bacteria Change after Exposure to Silver Nanoformulations: Analysis of the Genome and Outer Membrane Proteome

**DOI:** 10.3390/pathogens10070817

**Published:** 2021-06-29

**Authors:** Anna Kędziora, Mateusz Speruda, Maciej Wernecki, Bartłomiej Dudek, Katarzyna Kapczynska, Eva Krzyżewska, Jacek Rybka, Gabriela Bugla-Płoskońska

**Affiliations:** 1Department of Microbiology, Faculty of Biological Sciences, University of Wroclaw, 51-148 Wroclaw, Poland; mateusz.speruda@uwr.edu.pl (M.S.); maciej.wernecki@uwr.edu.pl (M.W.); bartlomiej.dudek@uwr.edu.pl (B.D.); 2Department of Immunology of Infectious Diseases, Hirszfeld Institute of Immunology and Experimental Therapy, Polish Academy of Sciences, 53-114 Wroclaw, Poland; katarzyna.kapczynska@hirszfeld.pl (K.K.); eva.krzyzewska@hirszfeld.pl (E.K.); jacek.rybka@hirszfeld.pl (J.R.)

**Keywords:** bacteria, pathogens, *Escherichia coli*, genome analysis, mutations, outer membrane proteins, OMP, silver nanoformulations, SNF

## Abstract

Objective: the main purpose of this work was to compare the genetic and phenotypic changes of *E. coli* treated with silver nanoformulations (*E. coli* BW25113 wt, *E. coli* BW25113 AgR, *E. coli* J53, *E. coli* ATCC 11229 wt, *E. coli* ATCC 11229 var. S2 and *E. coli* ATCC 11229 var. S7). Silver, as the metal with promising antibacterial properties, is currently widely used in medicine and the biomedical industry, in both ionic and nanoparticles forms. Silver nanoformulations are usually considered as one type of antibacterial agent, but their physical and chemical properties determine the way of interactions with the bacterial cell, the mode of action, and the bacterial cell response to silver. Methods: the changes in the bacterial genome, resulting from the treatment of bacteria with various silver nanoformulations, were verified by analyzing of genes (selected with mutfunc) and their conservative and non-conservative mutations selected with BLOSUM62. The phenotype was verified using an outer membrane proteome analysis (OMP isolation, 2-DE electrophoresis, and MS protein identification). Results: the variety of genetic and phenotypic changes in *E. coli* strains depends on the type of silver used for bacteria treatment. The most changes were identified in *E. coli* ATCC 11229 treated with silver nanoformulation signed as S2 (*E. coli* ATCC 11229 var. S2). We pinpointed 39 genes encoding proteins located in the outer membrane, 40 genes of their regulators, and 22 genes related to other outer membrane structures, such as flagellum, fimbria, lipopolysaccharide (LPS), or exopolysaccharide in this strain. Optical density of OmpC protein in *E. coli* electropherograms decreased after exposure to silver nanoformulation S7 (noticed in *E. coli* ATCC 11229 var. S7), and increased after treatment with the other silver nanoformulations (SNF) marked as S2 (noticed in *E. coli* ATCC 11229 var. S2). Increase of FliC protein optical density was identified in turn after Ag+ treatment (noticed in *E.coli* AgR). Conclusion: the results show that silver nanoformulations (SNF) exerts a selective pressure on bacteria causing both conservative and non-conservative mutations. The proteomic approach revealed that the levels of some proteins have changed after treatment with appropriate SNF.

## 1. Introduction

Since ancient times, silver has been known for its antiseptic properties. In the past, it was used as either silver ions or metallic silver. The development of nanotechnology has brought about new possibilities of using the antimicrobial potential of this metal. Among numerous metals with antibacterial properties, silver is in the spotlight of nanotechnologists due to its promising possibilities of use, both in and outside of the clinic. Over the last years, a significant increase in the production of silver nanomaterials has been observed, towards different purposes and with various methods being used, which is connected with a high diversity of silver nanomaterials in terms of size, shape, surface charge, and composition. The physical and chemical properties, frequently resulting from the type of synthesis, determine the interaction with bacterial cells, the mode of action, and bacteria’s cell response to silver and other antibacterial agents (such as antibiotics and biocides) [1,2,3]. The proposed mode of action of silver on bacterial cells has been previously described [2,4]. Direct interactions with bacterial cell structures and a physical impact on the cell envelope (outer membrane, peptidoglycan, or cell membrane) are the basic methods of cell disruption by forming gaps in the cell membrane or the inhibition of biochemical pathways. One of the proposed mechanisms is the uptake of silver into the cell by outer membrane proteins (OMP), such as OmpC and/or OmpF. Then, Ag atoms can interact with internal biomolecules, such as proteins and nucleic acids (DNA or RNA) [5,6,7,8]. Randall et al. [8] have confirmed that a bacterial cell deprived of OmpC and OmpF becomes more resistant to silver ions. A most important mode of action of silver is production of reactive oxygen species (ROS) that destroy cell components causing rearrangement in the cell envelope and changes in the biochemical pathways. A separate type of interaction with a bacterial cell is the intercalation with the genophore resulting in the inhibition of cell division or introduction of changes to the genetic material [2,5,8]. The substantial changes in the sensitivity of some bacterial strains to particular silver nanoformulations (SNF), and some antibiotics after a long-term treatment with SNF, as the effect of adaptation of the bacteria cell to the environmental stress or mutational changes, have been observed [1]. There are a number of papers speculating about and confirming the direct cytotoxicity of silver nanoparticles [4,5,8,9,10], but the genetic basis for the changes still remains unconfirmed. For a deeper explanation changes in the genes encoding OMP, flagella, fimbria, lipopolysaccharide (LPS), and exopolysaccharide by assigning the BLOSUM62 score [11,12], and based on an analysis of the mutfunc mutation database [13], has been showed. To our knowledge, this is the first publication where such a detailed analysis of the genome after bacteria treatment with selected SNF is shown. The *E. coli* cell phenotype using the outer membrane proteome analysis has also been included.

## 2. Results

### 2.1. Genome Analysis

Genes encoding outer membrane proteins carrying conservative and non-conservative mutations in *E. coli* ATCC 11229 var. S2, obtained using NGS genetic analysis, and grouped according to their involvement in molecular functions are listed in Table 1. They are related to transmembrane transport (channels, siderophore transporter and others), peptidoglycan (penicillin-binding protein activator, Braun’s lipoprotein, membrane-bound lysozyme inhibitor, membrane-bound lytic murein transglycosylase B and murein hydrolase activator), lipids, intermembrane phospholipid transport system lipoprotein and metalloprotease) and others (outer membrane protein assembly factors, cellulose synthase operon protein C, bacteriophage adsorption protein A, poly-beta-1;6-N-acetyl-D-glucosamine N-deacetylase, outer membrane lipoprotein, trans-aconitate 2-methyltransferase). Among 145 *E. coli* genes encoding outer membrane associated proteins, we have detected 39 genes with a total of 94 missense mutations, among which 26 were non-conservative mutations (BLOSUM62 criterion) and 7 mutations were selected by mutfunc as impactful (altered amino acids were present in the conservative region of the protein or alterations could potentially destabilize the protein). All the proteins, along with the list of conservative and non-conservative mutations, are summarized in Appendix A. Products of these selected genes are responsible for the bacterial cell structure, transport, secretion, adhesion, and adsorption. Among the OMP, the following porins: OmpC, OmpF, OmpG, and OmpN were pinpointed with conservative or non-conservative mutations (Appendix A). The genes encoding proteins, which are related to OM and other outer membrane structures, such as flagellum, fimbria, LPS, and exopolysaccharide, were analyzed in a different way and the genes selected with mutfunc together with the list of conservative mutations detected with BLOSUM62 are summarized in Appendix A, in total, 22 genes. Selected genes are mainly responsible for the structure of cellular flagellum, transport, oxidation stress, and respiratory cell. Moreover, regulators of genes identified in Appendix A were pinpointed (40 genes listed in Appendix A), in the list of conservative and non-conservative mutations. High number of mutations were identified in this case. Impactful mutations selected with mutfunc and accompanied by predicted effects of mutations for proteins were pinpointed in some transmembrane transporter activity *fimD, ompN, yehB, fhuE,* and *pgaA,* and others, such as *yceB* (uncharacterized lipoprotein).

### 2.2. Proteome Analysis

The proteomic analysis of the bacterial strains has also revealed significant differences. The comparison of electropherograms of the outer membrane proteome of *Escherichia coli* BW25113 and its silver-resistant mutant (*E. coli* AgR), shows several differences, which emerge after prolonged exposure of the wild type strain to Ag^+^. The protein spots with the most conspicuous alterations between strains were subjected to protein identification using mass spectrometry (Appendix A). All spots selected for analysis are clearly marked in Figure 1. The results showed that some differences did not solely concern a fraction of OMP, probably as a result from the isolation method, but were also recognized as cytosolic or inner membrane molecules with a crucial function, i.e., in the cellular response to oxidative stress, transport of macromolecules, aerobic/anaerobic metabolism, or the maintenance of heavy metal homeostasis. It could be helpful to understand the involvement of proteins from other cellular compartments and their role in the cellular response to the action of silver ions and nanoformulations. The presented 2-DE electropherograms (Figure 1) differ in the number of protein spots or in their optical density (OD). One of the noticeable differences among the electropherograms of *E. coli* BW25113 and its mutant AgR is connected to flagellin, a subunit protein involved in the formation of bacterial flagella. All proteins, irrespective of their position in the gel, were identified as FliC (spots 1–4, Figure 1). Spots 1 and 2 were not detected in the electropherogram of the wild type *E. coli* BW25113, while spots 3 and 4 differ in staining intensity. Those detected in the *E. coli* AgR mutant (especially spot no. 4) exhibit much higher OD that cannot be missed. The scattered location of flagellin could be explained by some modifications of FliC made shortly after the translation process, which have an impact on the molecular mass and pI value of proteins in each spot. In order to obtain additional information about the potential changes of the OMP of *E. coli*, we decided to expand the analysis by using another strain—*E. coli* J53, with resistance to Ag^+^ determined by the pMG101 plasmid (Figure 1, Table 2). In the protein profile of *E. coli* J53, none of the four previously mentioned spots was detected. Further analysis of *E. coli* BW25113 electropherograms made it possible to identify two changes concerning spots no. 7 and 8. They were recognized as D-galactose-binding periplasmic protein and superoxide dismutase. OD of both mentioned spots decreases in the electropherogram of *E. coli* BW25113 AgR, strain loss of the OmpC/F porins and derepression of the CusCFBA efflux transporter (Figure 1, Table 2). The spot identified as D-galactose-binding protein seems to have a higher OD in *E. coli* J53 than the corresponding spot recognized in *E. coli* BW25113 AgR. Furthermore, the OD value of the aforementioned *E. coli* J53 spot may be similar in terms of intensity to its wild type counterpart. In the electropherograms of *E. coli* BW25113 and its AgR mutant, there were two spots identified as aldehyde reductase YahK (no. 9) and protein CutC (no. 10). The aforedescribed spots could not be found in the electropherogram of *E. coli* J53. Other differences between the analyzed electropherograms concern two spots identified as chaperone protein DnaK (no. 5) and isocitrate lyase (no. 6). These spots were detected in both BW25113 strains (wt and AgR) with no differences between them. When compared to the protein profile of *E. coli* J53, they seem to be more distinctive, because the OD of both detected spots notably decreases in relation to the electropherograms of BW25113 strains. The last analyzed change was connected with superoxide dismutase (no. 8). Its intensity in the gels of *E. coli* AgR and J53 were much lower than the OD of the wild type corresponding spot (*E. coli* BW25113).

The next step of our research was carried out on *E. coli* ATCC 11229 strains, wild type and both variants (Figure 1). In case of both variants of *E. coli* ATCC 11229 var. S2 and S7, changes in the outer membrane structures were identified that affected the bacterial susceptibility to the tested nanomaterials and made *E. coli* cells more resistant to those silver nanoformulations (S2 and S7) [1]. Changes observed in 2-DE electropherograms were distinctive for each of the obtained variants (strain with silver-driven differences) and concerned the absence of selected spots or fluctuations of their OD. As in the case of the previously described *E.coli* BW25113, the group of the identified spots contained proteins with different localization (cytosol, periplasm, OM) and function.

The concentration of OmpC (no.1) decreased in the case of *E. coli* ATCC 11229 var. S7 in comparison with its wild type. The opposite situation was observed in the electropherogram of *E. coli* ATCC 11229 var. S2, where the OD of the aforementioned spot was much higher. A more significant decrease of OD (regarding the wild type) was observed in another spots (no. 2) of glutaredoxin-4 present on the gels of both variants. The results of the identification showed that spots no. 5 and 6 were actually the same protein—OPPA. It seems that spot no. 6 (present only in the electropherogram of *E. coli* ATCC 11229 var. S2) is probably a molecule that had undergone some post-transcriptional modifications. In *E. coli* ATCC 11229 var. S7, OD of spot no. 5 was higher compared to the wild type and S2. Nevertheless, in the case of the second variant mentioned above, the OPPA was represented by two spots; therefore, the final amount of this protein in both variant samples could be the same as in var. S7. Spot no. 3 (Figure 1) was recognized as D-galactose-binding periplasmic protein. The same structure was identified in the electropherograms of *E. coli* BW25113. The spot occurred only in the wild type and *E. coli* ATCC 11229 var. S7, but an increase of its OD could be observed only in the gel of the variant (Figure 1). The same situation was observed in the case of protein no. 7, identified as a thiosulfate-binding protein. The last analyzed difference between *E. coli* ATCC 11229 strains concerned spot no. 4, recognized as malate dehydrogenase (found only in *E. coli* ATCC 11229 var. S7). A correlation was discovered between the genetic changes and proteins detected in 2-DE, and referred to: (i) the subunit protein, which polymerizes to form the filaments of bacterial flagella (extracellular component)—*fliC* (conservative mutation) in *E. coli* BW25113 AgR; (ii) the active transport of galactose and glucose; (iii) chemotaxis towards the two sugars by interacting with the trg chemoreceptor—*mglB* (non-conservative mutation) in *E. coli* ATCC 11229 var. S7 (Table 3). Moreover, conservative mutation was detected in a component of the oligopeptide permease, a binding protein-dependent transport system, involved in the binding of peptides up to five amino acids long with high affinity to the—*appA* gene in *E. coli* ATCC 11229 var. S2 and S7 (Table 3). It is worth emphasizing that conservative mutation was also found in *rpoD*—initiation factors that promote the attachment of RNA polymerase to specific initiation sites and then its release. This factor is the primary sigma factor during exponential growth. Preferentially transcribed genes are associated with fast growth and they include ribosomal operons, other protein-synthesis related genes, rRNA- and tRNA-encoding genes, and *prfB* (directing to the termination of translation in response to the peptide chain termination codons UGA and UAA).

## 3. Discussion

Much effort has been made to explain the molecular mode of the antibacterial action of silver nanomaterials, and the observations made are inconsistent. On the basis of published results, it can be speculated that the antibacterial mechanism of action strongly depends on the physical and chemical properties of the used silver formulation probably such as size, shape, charge, composition, surface [1,2,4,9,10]. Moreover, after repeated, prolonged treatment with Ag^+^ or various types of silver nanoparticles, phenotypic as well as genetic changes in the bacterial cell have been observed [1,4,8]. Observation of the bacterial structures indicated the strong interaction of SNF with the cell wall [1,4]. Wen-Ru Li et al. [14] have shown that silver nanoparticles (with different sizes, 5 nm, 20 nm, respectively) can cause severe damage to bacteria cell, but they have not made a differentiation between silver ions and silver nanoparticles. Yan et al. [15] have confirmed that both tested silver forms (ions and nanoparticles) can penetrate into the bacterial cell. They have investigated the molecular mechanisms of the antimicrobial activity of silver nanoparticles in *Pseudomonas aeruginosa* using the proteomic approach and have suggested that the interference with cell-membrane functions and generation of intracellular reactive oxygen species (ROS) are the main pathways for the antibacterial activity of silver nanoparticles and silver ions. The differences between antibacterial mode of action in case of the two kinds of silver (Ag^+^ and silver nanoparticles) have also been indicated by the others and genotoxicity consequences have been initially performed [4,8,14]. Anuj et al. [16] have suggested that nanosilver can modify the membrane integrity of *E. coli* in addition to the obstruction of the activity of efflux pumps. Silver nanoparticles may be localized inside the *E. coli* cell membrane or they may completely separate the cell membrane causing membrane damage [16]. Our latest results suggest that multiple OMP proteins are responsible for uptake of silver ions and silver nanoformulations [4]. As we pinpointed, SNF were more efficacious against all tested bacterial strains than silver ions, and this was confirmed with computational methods: weaker interactions of Ag^0^ with amino acids of inner layers of both investigated proteins allow Ag^0^ to “slide” inside the cell more effectively—with a lower energy barrier in comparison to Ag^+^ [4]. However, according to Lok et al. [17] mode of action of silver ions was similar to spherical nano-Ag (average diameter 9.3 nm), but nano-Ag was found as efficacy at lower concentration than silver ions. Using the proteomic approach, we showed the decrease of OmpC protein expression in *E. coli* ATCC 11229 after exposure to SNF S7, and its increase after the sample treatment with S2 silver (this was observed only in *E. coli* ATCC 11229, cf. Figure 1, Table 3). In contrast to *E. coli* AgR strains carrying mutations after silver ions treatment [8], no changes in *ompR* was identified in case of *E. coli* ATCC var. S2. However, some conservative mutations in *ompC* and non-conservative mutations in *ompF* were separately noticed. Observed mutation in the *ompC* gene could have influenced the *ompC* gene overexpression and uptake the biocides to bacteria cell. In the *cusS* gene, different mutation was also noticed by us in *E. coli* ATCC var. S2 than in *E. coli* AgR obtained by Randall et al. [8]: Thr81Ala, Asn117Asp, Thr118Ser in contrast to Ile213Ser, Ala312Glu and Arg377His, respectively.

Anuj et al. [16] have postulated that O-antigen (part of the LPS) in the *E. coli* strain may also be responsible for the interaction of silver nanoparticles with bacterial cell, so strains with mutations in this part of cell structure may be less or more sensitive to silver nanoformulations. Besides O-antigen, flagella is considered as part of a bacterial structure responsible for increase to silver nanoparticles resistance. Wen-Ru Li et al. [14] have observed that the flagella of the *E. coli* strain have been damaged or even eliminated, finally causing impairment of the cell movement. The upregulation of the FliC protein has also been observed in the bacterial cell after treatment with silver ions and nanoparticles by Yan et al [15]. According to Panáček et al. [10], the flagella of *E. coli* cause aggregation of silver nanoparticles without any genetic changes. In our studies, OD of FliC spot increased after treatment with Ag^+^ in case of *E. coli* BW25113. We identified the conservative mutations in the regulatory genes of *fliC*, flagellar basal-body P-ring formation protein (flgA) and a regulator of cell motility (fliZ) in *E. coli* ATCC var. S2 strain (Appendix A). Moreover, the upregulation of *fliC* in the proteinogram of the endogenously silver-resistant strain *(E. coli* AgR) was observed by us, while the model of an exogenously silver-resistant strain, *E. coli* J53, exhibited downregulation of this protein (Figure 1). It is interesting that *E. coli* ATCC 11229 strain and its variants (S2 and S7), besides of numerous mutations indicated in *E. coli* ATCC 11229 var. S2, still stay sensitive to antibiotics, while other Gram-negative (such as *E. coli, Klebsiella pneumoniae* or *Enterobacter cloacae*) and Gram-positive bacteria (*Staphylococcus aureus*) bacteria strains stay the same or become resistant to antibiotics [4].

The consequences of the phenotypic (including membrane rearrangements) and genetic changes may alter the sensitivity of bacteria to biocides and antibiotics (depending on the properties of the applied silver type) after repeated treatment with silver ions or nanoformulations [1,16]. The implications of those mutations, and how those mutations are correlated with the silver nanoformulation treatment, is not clear now. The importance of the observed mutations and their correlation with the silver nanoformulations need more explanation.

## 4. Materials and Methods

### 4.1. Materials

#### 4.1.1. Strains

The following bacteria strains were subjected to proteomic and genetic analysis to assess changes induced by the silver treatment: *E. coli* BW25113 wt (wild type) and its mutant *E. coli* BW25113 AgR, *E. coli* J53, *E. coli* ATCC 11229 wt (wild type) and its variants: *E. coli* ATCC 11229 var. S2, and *E. coli* ATCC 11229 var. S7 (they are described in details in Table 4) [1,8,18,19]. They were store in −70 °C (*Revco*) after selection as described previously [1,8]. We compared the proteome changes in all of the tested strains, additionally one of the strains: *E. coli* ATCC 11229 var. S2 was selected for a detailed genetic analysis according to the number of the selected mutations (general genetic information was mentioned in our previous study [1]). It is worth emphasizing that in this case attempts to obtain variants of *E. coli* BW25113 resistant to S2 and S7 failed, as this strain has remained sensitive to those silver nanoformulations samples during our experiment.

#### 4.1.2. Reagents

##### Genomic DNA Isolation

Culture medium—Lysogeny Broth Lennox (LB) (Sigma-Aldrich, Saint Louis, MO, USA); Genomic Mini Kit (A&A Biotechnology, Gdańsk, Poland).

##### OMP Isolation

Culture medium—Lysogeny Broth Lennox (LB) (Sigma-Aldrich, Saint Louis, USA); buffer A: 1M sodium acetate (POCH), 1mM β-mercaptoethanol (Merck, Darmstadt, Germany), ultra-pure water; buffer B: 5% (*w*/*v*) Zwittergent Z 3–14^®^ (Calbiochem, San Diego, CA, USA), 0.5 M CaCl_2_ (POCH, Gliwice, Poland), ultra-pure water; buffer C: 50 mM Trizma-Base (Sigma-Aldrich, Saint Louis, USA), 0.05% (*w*/*v*) Zwittergent Z 3-14^®^ (Calbiochem, San Diego, USA), 10 mM EDTA (Sigma-Aldrich, Saint Louis, USA) (pH 8.0); 96% (*v*/*v*) ethanol (Merck, Darmstadt, Germany), ultra-pure water; BCA Protein Assay Kit (Thermo Scientific, Waltham, MA, USA).

##### Two-Dimensional Gel Electrophoresis

Immobilized gradient IPG strips (pH 4–7, 7 cm) (Bio-Rad, Hercules, CA, USA), ReadyPrep™ 2-D Cleanup Kit (Bio-Rad, CA, USA), Mini-PROTEAN Tetra Cell System (Bio-Rad), agarose (Bio-Rad, CA, USA), Coomassie Brilliant Blue (Bio-Rad, CA, USA), polyacrylamide gels: 30% acrylamide/bis-acrylamide solution (Bio-Shop, Burlington, Canada), TRIS (Sigma-Aldrich, Saint Louis, USA), HCl (POCH, Gliwice, Poland), 10% (*w*/*v*) sodium dodecyl sulfate (Sigma-Aldrich, Saint Louis, USA), 10% (*w*/*v*) ammonium persulfate (Bio-Shop, Burlington, ON, Canada), TEMED (Bio-Shop, Burlington, Canada), ultra-pure water.

##### In-Gel Protein Digestion and MS Protein Identification

Ultra-pure water and LC–MS grade solvents were used for protein digestion and MS experiments. De-staining buffer: 100 mM NH_4_HCO_3_ (Merck, Darmstadt, Germany), acetonitrile (Merck, Darmstadt, Germany), 1:1 *v*/*v*; reduction buffer: 10 mM dithiothreitol (Sigma-Aldrich, Saint Louis, USA) in 100 mM NH_4_HCO_3_ (Merck, Darmstadt, Germany), alkylation buffer: 55 mM iodoacetamide (Sigma-Aldrich, Saint Louis, USA) in 100 mM NH_4_HCO_3_ (Merck, Darmstadt, Germany), trypsin solution: 13 ng/μL trypsin (Promega, Madison, WI, USA) in 10 mM NH_4_HCO_3_ (Merck, Darmstadt, Germany) containing 10% (*v*/*v*) acetonitrile (Merck, Darmstadt, Germany); Pierce C18 pipette tips (Thermo-Scientific, Waltham, USA), elution buffer: 10 mg/mL of α-cyano-4-hydroxycinnamic acid (Bruker, Billerica, MA, USA) in acetonitrile: 0.1% TFA (Merck, Darmstadt, Germany) in H_2_O 7:3 *v*/*v*.

### 4.2. Methods

#### 4.2.1. Genome Analysis

Genomic DNA of the bacteria was isolated using Genomic Mini Kit (2 mL of overnight bacterial cultures in LB medium) (Biomaxima, Lublin, Poland). The purity and concentration of the product was measured with a nano spectrophotometer (Implen). Genomic libraries were prepared using NEBNext DNA Library Prep Master Mix Set for Illumina and sequencing was performed in Genomed (Warsaw, Poland) using Illumina MiSeq. NGS reads were preprocessed with Cutadapt 1.9.1 [20], assembled with spades [21], and contigs were rearranged with progressive Mauve in Mauve 2.4.0 [22,23], using the genome of *E. coli* K-12 substr. MG1655 (NC_000913.3) as reference. A mutations list was generated with snippy [24]. Membrane related genes containing mutations were extracted from the list on the basis of the results obtained from the UniProt KB database (keywords: flagellum (KW-0975), bacterial flagellum biogenesis (KW-1005), cell adhesion (KW-0130), exopolysaccharide synthesis (KW-0270), fimbrium biogenesis (KW-1029), fimbrium structural protein (KW-0281), flagellar rotation (KW-0283), lipopolysaccharide biosynthesis (KW-0448), membrane (KW-0472), and the organism *Escherichia coli* (strain K12) (83333)) and from the current *E. coli* Membranome database [25]. Non-synonymous mutations in those genes were assessed in two ways: by assigning the BLOSUM62 score [11,12], and using an analysis of the mutfunc mutation database [26]. Non-conservative mutations were selected as those with a negative BLOSUM62 score or were proposed directly by mutfunc. NGS reads are available in the NCBI SRA database (SRR9733699, SRR9733700, SRR9733697), and the assembled genomes in the NCBI Nucleotide database (VLTC00000000, VLTB00000000, VLTA00000000, and ASRI00000000).

#### 4.2.2. Proteome Analysis

OMP isolation was performed according to the Murphy and Bartos procedure with minor modifications [27,28]. Overnight culture of bacteria (LB medium, 37 °C, 18–24 h) was harvested (1500 g, 4 °C, 20 min) and suspended in 1.25 mL of buffer A. 11.25 mL of buffer B was added and the mixture was stirred (rt, 1 h). 3.13 mL of cold ethanol was added slowly in order to precipitate the nucleic acids and the mixture was centrifuged (17,000× *g*, 4 °C, 10 min). Then 46.75 mL of cold ethanol was added to the supernatant (17,000 g, 4 °C, 20 min). The pellet was dried, resuspended in 2.5 mL of buffer C and stirred (rt, 1 h). The solution was incubated at 4 °C overnight. OMP remained in the soluble fraction of the buffer and the insoluble material was removed by centrifugation (12,000× *g*, 4 °C, 10 min). BCA Protein Assay Kit was used for the total protein concentration measurement. ReadyPrep 2-D clean up kit (Bio-Rad) was used for sample preparation. Isoelectric focusing (IEF) was performed by a stepwise increase of voltage as follows: 250 V, 20 min; 4000 V, 120 min (linear) and 4000 V (rapid), until the total volt-hours reached 14 kVh. IPG strips were loaded onto the top of gel slabs using 0.5% agarose in the running buffer. Electrophoresis was carried out at 4 °C with constant power current (1 W) until the dye front reached the bottom of the slab [29,30]. Protein spots were visualized with Coomassie Brilliant Blue staining. PDQuest software (Bio-Rad) was used for protein spot pattern analysis. Protein spots selected for the mass spectrometry analysis were subjected to the in-gel tryptic digestion according to Shevchenko et al. [31] Mass spectrometry analysis using the MALDI TOF ultrafleXtreme instrument (Bruker Daltonics) was performed afterwards. The peptides were eluted directly on a MALDI plate using a solution of α-cyano-4-hydroxycinnamic acid as the matrix. Protein identification was accomplished with a bioinformatics platform (ProteinScape, Bruker Daltonics) and MASCOT (Matrix Science) as a search engine in protein sequence databases (NCBI, SwissProt).

## 5. Conclusions

Both *E. coli* wild types ATCC 11229 and *E. coli* BW25113 were treated with silver ions and silver nanoformulations (SNF), but *E. coli* ATCC 11229 has changed sensitivity mainly after treatment with SNF S2, the change in sensitivity was less for SNF S7, and no change of sensitivity was observed for Ag^+^, while *E. coli* BW25113 has not changed after exposure to SNF S2 and SNF S7, but showed derivations after silver ions treatment. Silver nanoformulations exert a selective pressure on bacterial cells, causing both conservative and non-conservative mutations, and/or phenotypical changes in different way than silver ions. A genetic analysis by the whole-genome sequencing provided a better understanding of the interactions between silver nanoformulations and *E. coli* strains. The following genes were selected with mutfunc and analyzed by Blosum62 as those with conservative and non-conservative mutations: encoding proteins located in the outer membrane (including OmpC, OmpF, OmpG, and OmpN) and their regulators, genes related to OM and other outer membrane structures, such as flagellum, fimbria, LPS, or exopolysaccharide. Using the proteomic approach with protein isolation and 2-DE experiment, we showed that the optical density changed for some protein spots in 2D electropherograms, such as OmpC or FliC, isocitrate lyase AceA, chaperone protein DnaK, D-galactose binding protein MglB, thiosulfate-binding protein CysP, malate dehydrogenase Mgh, glutaredoxin-4 GrxD, periplasmic oligopeptide-binding protein OppA, and OmpC, depending on the silver form used for treatment. The molecular mechanism of the antibacterial activity of silver and molecular changes in bacterial cells strongly depend on the physical and chemical properties of the tested SNF form. At this time, it is difficult to conclude what physico-chemical properties determine antibacterial cytotoxicity and genotoxicity. Therefore, the mode of action of antibacterial SNF is apparently much more complex and the phenomenon of bacterial resistance to silver requires further and deeper studies.

## Figures and Tables

**Figure 1 pathogens-10-00817-f001:**
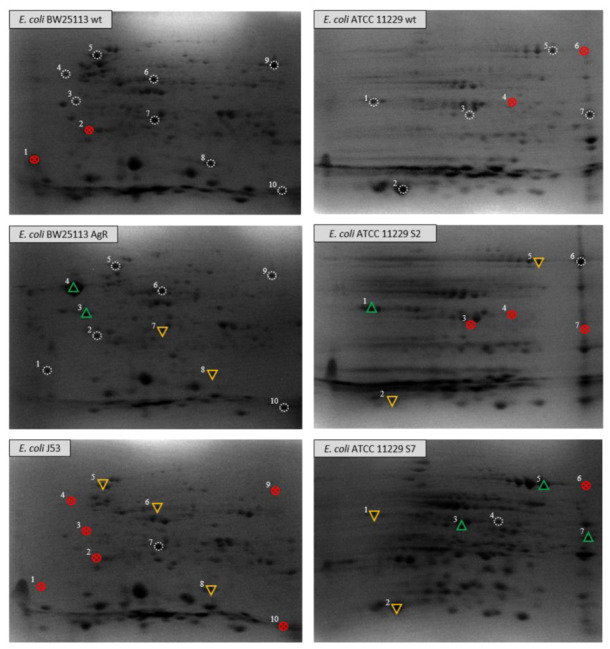
Comparative analysis of two-dimensional electropherograms of *Escherichia coli* BW25113 and its mutant AgR resistant to silver ions and *Escherichia coli* ATCC 11229 and its variants: S2 and S7. Legend: 
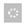
 reference spot; 
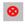
 selected spots not found in tested sample; 
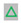
 spot with increased optical density; 
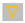
 spot with decreased optical density.

**Table 1 pathogens-10-00817-t001:** Genes encoding outer membrane proteins carrying mutations, grouped according to their involvement in molecular functions.

Function of the Gene Product		Gene	Product Description	Mutations	Predicted Effects
Transmembrane transporter activity	Channel	*cusC*	Cation efflux system protein CusC	Thr81Ala, Asn117Asp, Thr118Ser	-
		*fimD*	Outer membrane usher protein	Ser397Thr, Ile401Val, **Gly403Ala**, Glu410Lys	Alteration in a conserved region, destabilizing mutation
		*ompC*	Outer membrane porin C	Ile306Val Thr305Val	-
		*ompF*	Outer membrane porin F	Asp48Gly, Val51Glu, Lys60Met,	-
		*ompG*	Outer membrane porin G	Ala67Ser	-
		*ompN*	Outer membrane porin N	**Lys90Thr**	Destabilizing mutation
		*tsx*	Nucleoside-specific channel-forming protein	Ile53Leu	-
		*uidC*	Membrane-associated protein	Asn285Ile	-
		*yehB*	Outer membrane usher protein	Thr377Ser, Gln540His, Ile699Val	Alteration in a conserved region
		*yfaL*	Probable autotransporter	Thr879Ala, Pro920Pro, Pro921Thr, Ser922Pro, Thr919Asn, Ser1091Asn Glu1098Asp Thr1101Val	-
	Siderophore transporter	*cirA*	Colicin I receptor	Ile174Val	-
		*fecA*	Fe(3+) dicitrate transport protein	Ile56Phe, Val350Leu, Ile397Val, Gly600Asp, Val605Ile, Leu651Phe, Ser684Ala, Met693Ile	-
		*fepA*	Ferrienterobactin receptor	Thr356Ala	-
		*fhuA*	Ferrichrome outer membrane transporter/phage receptor	Ser110Ala	-
		*fhuE*	Outer-membrane receptor for Fe(III)-coprogen; Fe(III)-ferrioxamine B and Fe(III)-rhodotorulic acid	**Ser61Asn**	destabilizing mutation
		*fiu*	Catecholate siderophore receptor	Thr493Ala	-
	Others	*gspD*	Putative type II secretion system protein D	Gln617Leu, Val627Ile	-
		*pgaA*	Poly-beta-1;6-N-acetyl-D-glucosamine (PGA) export protein	Gln11Lys, Ile18Leu, Val26Ala, Ile87Val, Ser90Arg, Ile106Val, **Pro129Ser**, Thr150Ser, **Pro451His**, Phe599Val	Alteration in conserved regions
Peptidoglycan-related		*lpoA*	Penicillin-binding protein activator	Ala96Thr, Val106Ala, Ala292Val	-
		*lpoB*	Penicillin-binding protein activator	Pro71Leu	-
		*lpp*	Major outer membrane lipoprotein (Braun lipoprotein)	Val26Ile	-
		*mliC*	Membrane-bound lysozyme inhibitor of C-type lysozyme	His25Arg, Ala29Asp	-
		*mltB*	Membrane-bound lytic murein transglycosylase B	Val49Met, Asp64Glu, Lys324Arg	-
		*nlpD*	Murein hydrolase activator	Ala228Thr	-
Lipid-related		*blc*	Outer membrane lipoprotein	Gly84Glu	-
		*lolB*	Outer membrane lipoprotein	Ala115Ser	-
		*mlaA*	Intermembrane phospholipid transport system lipoprotein	Gly168Ser	-
		*ycaL*	Metalloprotease	Ser158Arg	-
Various	Bam	*bamB*	Outer membrane protein assembly factor	Ser96Asn, Ser335Gly	-
		*bamC*	Outer membrane protein assembly factor	Asp287Glu, Gln289His	-
	Others	*bcsC*	Cellulose synthase operon protein C	Val65Ile, Pro110Ser, Ala558Gly, Ala775Thr	-
		*nfrA*	Bacteriophage adsorption protein A	Ala98Asp, Ala115Asp, Ile784Leu	-
		*pgaB*	Poly-beta-1;6-N-acetyl-D-glucosamine N-deacetylase (PGA N-deacetylase)	Leu575Met	-
		*slyB*	Outer membrane lipoprotein	Val78Ile	-
		*tam*	Trans-aconitate 2-methyltransferase	Asn121Ser, Gln123Leu, Ser158Pro, Ile162Val, Ala210Thr Leu216His	-
		*yajI*	Uncharacterized lipoprotein	Glu164Asp, Asp167Gly	-
		*yceB*	Uncharacterized lipoprotein	**Glu126Gly**	Alteration in a conserved region
		*yfeY*	Uncharacterized protein	Gly142Ser, Arg167Ser	-

Non-conservative mutations selected with BLOSUM62 criterion are italicized. Impactful mutations selected with mutfunc are marked in bold and accompanied by predicted effects of mutations for proteins.

**Table 2 pathogens-10-00817-t002:** Identified proteins from the selected spots of electropherograms of *E. coli* BW25113, its mutant *E. coli* BW25113 AgR and *E. coli* J53 as reference.

Spots In Figure 1	Identified Proteins (Encoding Gene)	Data	Strains	Function
1–4	Flagellin FliC (*flyC*)	MW = 51,265 pI = 4.5	*E. coli* BW25113 AgR only	Subunit protein which polymerizes to form the filaments of bacterial flagellum (extracellular component)
5	Chaperone protein DnaK (*dnaK*)	MW = 69,072 pI = 4.83	*E. coli* BW25113 wt	Essential role in the initiation of phage lambda DNA replication, involved in chromosomal DNA replication, participates actively in the response to hyperosmotic shock
6	Isocitrate lyase AceA (*aceA*)	MW = 47,782 pI = 5.16	*E. coli* BW25113 AgR only	Involved in the metabolic adaptation in response to environmental changes; catalyzes the reversible formation of succinate and glyoxylate from isocitrate, a key step of the glyoxylate cycle, which operates as an anaplerotic route for replenishing the tricarboxylic acid cycle during growth on fatty acid substrates, metal binding
7	D-galactose binding protein MglB (*mglB*)	MW = 35,690 pI = 5.68	*E. coli* BW25113 AgR only	Protein involved in the active transport of galactose and glucose. It plays a role in the chemotaxis towards the two sugars by interacting with the *trg* chemoreceptor
8	Superoxide dismutase SodF, SODF, SOD2, FeSOD I (*sodF*)	MW = 21,311 pI = 5.58	*E. coli* BW25113 wt	Destroys superoxide anion radicals which are normally produced within the cells and which are toxic to biological systems
9	Aldehyde reductase YahK (*yahK*)	MW = 37,954 pI = 5.80	*E. coli* BW25113 wt	Uncharacterized zinc-type alcohol dehydrogenase
10	Copper homeostasis protein CutC (*cutC*)	MW = 26,623 pI = 5.75	*E. coli* BW25113 wt and AgR	Control of copper homeostasis, copper ions binding

MW [Da]—molecular mass, pI—isoelectric point.

**Table 3 pathogens-10-00817-t003:** Identified proteins from the selected spots of electropherograms of *E. coli* ATCC 11229 and its variants *E. coli* ATCC 11229 var. S2 and S7.

Spots in Figure 1	Identified Protein (Encoding Gene)	Data	Strains	Function (https://www.uniprot.org/ accessed on 12 November 2020)
1	OmpC (*ompC*)	M = 40,343, pI = 4.58	*E. coli* ATCC 11229 wt only	Forms pores that allow passive diffusion of small molecules across the outer membrane. Microbial infection: supports colicin E5 entry in the absence of its major receptor OmpF, A mixed OmpC-OmpF heterotrimer is the outer membrane receptor for toxin CdiA-EC536; polymorphisms in extracellular loops 4 and 5 of OmpC confer susceptibility to CdiA-EC536-mediated toxicity.
2	Glutaredoxin-4 GrxD (*grxD*)	M = 13,044, pI = 4.75	*E. coli* ATCC 11229 wt only	Monothiol glutaredoxin involved in the biogenesis of iron-sulfur clusters.
3	D-galactose-binding periplasmic protein MglB (*mglB*)	M = 35,690, pI = 5.68	*E. coli* ATCC 11229 var. S7	Protein involved in the active transport of galactose and glucose. It plays a role in the chemotaxis towards the two sugars by interacting with the trg chemoreceptor.
4	Malate dehydrogenase Mgh (*mdh*)	M = 32,317, pI = 5.61	*E. coli* ATCC 11229 var. S7	Catalyzes the reversible oxidation of malate to oxaloacetate.
5–6	Periplasmic oligopeptide-binding protein OppA (*oppA*)	M = 60,977, pI = 6.05	*E. coli* ATCC 11229 var. S2 and S7	Component of the oligopeptide permease, a binding protein-dependent transport system, it binds peptides up to five amino acids long with high affinity.
7	Thiosulfate-binding protein CysP (*cysP*)	M = 37,591, pI = 6.81	*E. coli* ATCC 11229 wt only	Part of the ABC transporter complex CysAWTP involved in sulfate/thiosulfate import. This protein specifically binds thiosulfate and is involved in its transmembrane transport.

MW [Da]—molecular mass, pI—isoelectric point.

**Table 4 pathogens-10-00817-t004:** Bacteria strains used in this work.

Bacteria Strains		Genome Analysis	Proteome Analysis
Wild Type Strains and Its Variants	Description		
*E. coli* BW25113 wt	Wild type strain	[8] (Reference strain)	This work
*E. coli* BW25113 AgR (full name: *E. coli* BW25113 ompRG596AcusSG1130A)	Variant of *E. coli* BW25113 wt resistant to silver ions (Ag^+^) due to mutations, in ompR and cusS, respectively, conferred loss of the OmpC/F porins and derepression of the CusCFBA efflux transporter	[8]	This work
*E. coli* J53	Model organism with pMG101 plasmid encoding *sil* genes determining the resistance to silver ions	[8] (Reference strain with exogenous resistance to silver ions)	This work
*E. coli* ATCC 11229 wt	Wild type strain treated with different silver (ions and nanoformulations)	This work (reference strain)	This work
*E. coli* ATCC 11229 var. S2	Variant of *E. coli* ATCC 11229 wt with decreased sensitivity to SNF S2 * (after S2 treatment)	This work	This work
*E. coli* ATCC 11229 var. S7	Variant of *E. coli* ATCC 11229 wt with decreased sensitivity to SNF S7 * (after S7 treatment)	Small number of mutations analyzed in previous work [1]	This work

* Legend: silver nanoformulation signed as S2 refers to titanium dioxide doped with silver nanoparticles, while silver nanoformulation signed as S7 refers to water colloid of silver [18,19].

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
