# Peer review of "How Bacteria Change after Exposure to Silver Nanoformulations: Analysis of the Genome and Outer Membrane Proteome"

_pathogens, 2021, doi:10.3390/pathogens10070817_

Round 1

Reviewer 1 Report

The objective of the manuscript was to compare the genetic and phenotypic changes induced by silver nanoformulations on bacteria, using several strains of E. coli as a model.

This study is in the scope of the journal. This manuscript has an interesting point of view on the use of silver nanoparticles as a biocide. However, the authors wrote the article in an overly personalistic way. Expressions such as "The proposed mode of action of silver on bacterial cells has been analysed by us previously." (Ln. 53, p. 2) may perhaps be replaced by: The proposed mode of action of silver on bacterial cells has been previously described. It is clear that the work has been developed by the same research team, but it is less personal and more neutral.

Other examples:

Ln. 71, Pg. 2: “For a deeper explanation we show here changes in the genes encoding OMP…”. Perhaps may be replaced by: For a deeper explanation changes in the genes encoding OMP…has been showed.

Ln. 75 Pg. 2: “We have also studied the E. coli cell phenotype using the outer membrane proteome analysis”. Perhaps may be replaced by: The E. coli cell phenotype using the outer membrane proteome analysis has been included.

Please rewrite the rest of the manuscript accordingly.

Ln. 78 and followings. Pg. 2. “In the present work several E. coli strains (wild types and variants), which were previously treated with silver ions or silver nanoformulations1,7, were subjected to proteomic and genetic analysis to assess changes induced by the silver treatment. We had compared the proteome changes in all of the tested strains, additionally one of the strains: E. coli ATCC 11229 var. S2 was selected for a detailed genetic analysis according to the number of the selected mutations (general genetic information was mentioned in our previous study 1). It is worth emphasizing that in this case attempts to obtain variants of E. coli BW 25113 resistant to S2 and S7 failed, as this strain has remained sensitive to those silver nanoformulations samples during our experiment”.

This is material and methods, not results, in fact it is a justification for section 4.1.1. Change the section of this paragraph.

Figure 1, p. 9: If possible, a modification of the image quality would be recommended. Perhaps the authors can do a scan of the gels. Understandably, they probably don't have the original gel, but it doesn't look good.

This article is an interesting contribution to elucidate the action of silver nanoparticles.

Author Response

# Reviewer 1.

The objective of the manuscript was to compare the genetic and phenotypic changes induced by silver nanoformulations on bacteria, using several strains of E. coli as a model.

This study is in the scope of the journal. This manuscript has an interesting point of view on the use of silver nanoparticles as a biocide. However, the authors wrote the article in an overly personalistic way. Expressions such as "The proposed mode of action of silver on bacterial cells has been analysed by us previously." (Ln. 53, p. 2) may perhaps be replaced by: The proposed mode of action of silver on bacterial cells has been previously described. It is clear that the work has been developed by the same research team, but it is less personal and more neutral.

Other examples:

Ln. 71, Pg. 2: “For a deeper explanation we show here changes in the genes encoding OMP…”. Perhaps may be replaced by: For a deeper explanation changes in the genes encoding OMP…has been showed.

Ln. 75 Pg. 2: “We have also studied the E. coli cell phenotype using the outer membrane proteome analysis”. Perhaps may be replaced by: The E. coli cell phenotype using the outer membrane proteome analysis has been included.

Please rewrite the rest of the manuscript accordingly.

Answer: Thank you for your valuable comments and suggestions. It has been rewritten in more neutral way where it was possible and noticed.

Ln. 78 and followings. Pg. 2. “In the present work several E. coli strains (wild types and variants), which were previously treated with silver ions or silver nanoformulations1,7, were subjected to proteomic and genetic analysis to assess changes induced by the silver treatment. We had compared the proteome changes in all of the tested strains, additionally one of the strains: E. coli ATCC 11229 var. S2 was selected for a detailed genetic analysis according to the number of the selected mutations (general genetic information was mentioned in our previous study 1). It is worth emphasizing that in this case attempts to obtain variants of E. coli BW 25113 resistant to S2 and S7 failed, as this strain has remained sensitive to those silver nanoformulations samples during our experiment”.

This is material and methods, not results, in fact it is a justification for section 4.1.1. Change the section of this paragraph.

Answer: We had mixed filings in this case when we were writing the manuscript, so your comment is precious. It has been changed.

Figure 1, p. 9: If possible, a modification of the image quality would be recommended. Perhaps the authors can do a scan of the gels. Understandably, they probably don't have the original gel, but it doesn't look good.

Answer:The image has been changed but unfortunately, we’re not able to improve the quality of picture 1 any further. Photos of each electropherogram were taken by Gel Doc XR system and saved in .tif format, which was the best option for storing our images that we could choose at that time. Finally, Picture 1 was also saved as a .tif file.

Most changes were indicated in green in the main text.

This article is an interesting contribution to elucidate the action of silver nanoparticles.

Reviewer 2 Report

       In this manuscript, the authors have studied the genetic and phenotypic changes of E. coli (six different strains) treated with silver nanoformulations. Overall the study is carefully done and the manuscript is written with few exceptions ( see below). In my opinion, the manuscript is likely to be of interest to a diverse readership, including those interested in the antibiotic resistance mechanism.

There are a few suggestions/questions to authors which in my opinion will further improve the content of the manuscript:

Major issues:

  1. There is no doubt about the observations made by the author in terms of changes in the genome as well as Proteome. My big concern is the discussion section and explanation about the changes observed in the bacterial genome. The authors have just listed the conservative and non-conservative mutations that they have seen after sequencing. However, what is the implications of those mutations and how those mutations are correlated with the silver nanoformulations treatment is not clear? In my opinion, the author should attempt to get into more details discussing the importance of those mutations (whether they are gain of function/ loss of function or no change on function). In short importance of the observed mutations and their correlation with the silver, nanoformulations need more explanation. From a reader points of view, the current version of the manuscript seems to be just observations without gaining insight into the implications of such observations.
  2. The author should also try to check the effect of silver nanoformulations on gene knock out strains and see whether the knock out strains become mores sensitive or resistant? This need to be done with a subset of non-essential genes from list 2 in the manuscript.

Minor issues:

  1. The organism name should be italicized ( Line 33).
  2. Some spacing problems between words in manuscripts need further proofreading from the authors.( example- line 180).
  3. Figure 1 image needs to be changed. It will be better to provide the Coomassie-stained gel image rather than the Black/white image.
  4. Kindly include the mass-sec result data as supplementary information.

Author Response

# Reviewer 2.

In this manuscript, the authors have studied the genetic and phenotypic changes of E. coli (six different strains) treated with silver nanoformulations. Overall the study is carefully done and the manuscript is written with few exceptions ( see below). In my opinion, the manuscript is likely to be of interest to a diverse readership, including those interested in the antibiotic resistance mechanism.

There are a few suggestions/questions to authors which in my opinion will further improve the content of the manuscript:

Major issues:

  1. There is no doubt about the observations made by the author in terms of changes in the genome as well as Proteome. My big concern is the discussion section and explanation about the changes observed in the bacterial genome. The authors have just listed the conservative and non-conservative mutations that they have seen after sequencing. However, what is the implications of those mutations and how those mutations are correlated with the silver nanoformulations treatment is not clear? In my opinion, the author should attempt to get into more details discussing the importance of those mutations (whether they are gain of function/ loss of function or no change on function). In short importance of the observed mutations and their correlation with the silver, nanoformulations need more explanation. From a reader points of view, the current version of the manuscript seems to be just observations without gaining insight into the implications of such observations.

Answer: Thank you for this comments. We completely agree with you. When we were writing this manuscript we had mixed fillings with deeper interpretation. We felt that there would be too much speculations instead of facts if we wrote more about implication of those mutations. It would be difficult to predict the consequences. Therefore to avoid impression  “it was not confirmed in this study” we decided to write it as observation. We think that it could be a good idea to check transcriptome of those mutants and then we will be able to give deeper explanation. We have already planned such studies to performed them in the future. Impactful mutations selected with mutfunc were marked in bold and accompanied by predicted effects of mutations for proteins in table 2.

We added also short paragraph describing this issue. 

  1. The author should also try to check the effect of silver nanoformulations on gene knock out strains and see whether the knock out strains become more sensitive or resistant? This need to be done with a subset of non-essential genes from list 2 in the manuscript.

Answer: Thank you for your comments and suggestions. This is very good idea for further studies. We have a plan to check the effect with some genes listed in the manuscript.  

Minor issues:

  1. The organism name should be italicized ( Line 33).

Answer: It has been done

  1. Some spacing problems between words in manuscripts need further proofreading from the authors.( example- line 180).

Answer: It has been done

  1. Figure 1 image needs to be changed. It will be better to provide the Coomassie-stained gel image rather than the Black/white image.

Answer: Images of electropherograms taken by our Gel Doc XR system can be saved only as black and white. There’s no option given by the software to save and store the photos in its native, blue color. Nevertheless, we think that CBB as a staining method used in this part of our research was adequately highlighted in ‘Materials and methods’ subsection (and/or description of picture 1 presented in this article).

  1. Kindly include the mass-sec result data as supplementary information.

Answer: It has been done. All mass-sec results of each spot or E. coli BW 25113 and E. coli ATCC 11229 were added as Appendix.

Most changes were indicated with green in the main text.

Round 2

Reviewer 2 Report

I am satisfied with the response from the author and the incorporation of suggestions. Therefore no further comments.